# First Organic–Inorganic Hybrid Compounds Formed by Ge-V-O Clusters and Transition Metal Complexes of Aromatic Organic Ligands

**DOI:** 10.3390/molecules27144424

**Published:** 2022-07-11

**Authors:** Hai-Yang Guo, Hui Qi, Xiao Zhang, Xiao-Bing Cui

**Affiliations:** 1State Key Laboratory of Inorganic Synthesis and Preparative Chemistry and College of Chemistry, Jilin University, Changchun 130021, China; guohy@zjxu.edu.cn; 2College of Biological, Chemical Science and Engineering, Jiaxing University, Jiaxing 314001, China; 3The Second Hospital of Jilin University, Changchun 130021, China; qihui1977@sohu.com; 4MIIT Key Laboratory of Critical Materials Technology for New Energy Conversion and Storage, School of Chemistry and Chemical Engineering, Harbin Institute of Technology, Harbin 150001, China; zhangx@hit.edu.cn

**Keywords:** polyoxometalates, vanadogermanate, secondary transition metal substituted Ge-V-O clusters

## Abstract

Three compounds based on Ge-V-O clusters were hydrothermally synthesized and characterized by IR, UV-Vis, XRD, ESR, elemental analysis and X-ray crystal structural analysis. Both [Cd(phen)(en)]_2_[Cd_2_(phen)_2_V_12_O_40_Ge_8_(OH)_8_(H_2_O)]∙12.5H_2_O (**1**) and [Cd(DETA)]_2_[Cd(DETA)_2_]_0.5_[Cd_2_(phen)_2_V_12_O_41_Ge_8_(OH)_7_(0.5H_2_O)]∙7.5H_2_O (**2**) (1,10-phen = 1,10-phenanthroline, en = ethylenediamine, DETA = diethylenetriamine) are the first Ge-V-O cluster compounds containing aromatic organic ligands. Compound **1** is the first dimer of Ge-V-O clusters, which is linked by a double bridge of two [Cd(phen)(en)]^2+^. Compound **2** exhibits an unprecedented 1-D chain structure formed by Ge-V-O clusters and [Cd_2_(DETA)_2_]^4+^ transition metal complexes (TMCs). [Cd(en)_3_]{[Cd(η_2_-en)_2_]_3_[Cd(η_2_-en)(η_2_-μ_2_-en)(η_2_-en)Cd][Ge_6_V_15_O_48_(H_2_O)]}∙5.5H_2_O (**3**) is a novel 3-D structure which is constructed from [Ge_6_V_15_O_48_(H_2_O)]^12−^ and four different types of TMCs. We also synthesized [Zn_2_(enMe)_3_][Zn(enMe)]_2_[Zn(enMe)_2_(H_2_O)]_2_[Ge_6_V_15_O_48_(H_2_O)]∙3H_2_O (**4**) and [Cd(en)_2_]_2_{H_8_[Cd(en)]_2_Ge_8_V_12_O_48_(H_2_O)}∙6H_2_O (**5**) (enMe = 1,2-propanediamine), which have been reported previously. In addition, the catalytic properties of these five compounds for styrene epoxidation have been assessed.

## 1. Introduction

Several metallic materials have extensive uses, such as sensors, catalysis, fluids, regulated drug delivery and pigments [1,2,3,4,5]. Integration of certain metals to form polyoxometalates (POM) is a feasible and promising strategy for making new heteropolyoxometalates, which are of great interest due to their abundant structures and conceivable applications in magnetism, catalysis, medicine and electrochemistry [6,7,8,9,10,11,12,13,14,15]. Many different elements have been reported as compositions of heteropolyanions [16,17,18], and POMs containing different elements inspired an enormous amount of new research due to their range of intriguing applications [19,20,21,22,23,24]. Vanadium is of particular interest since it shows flexible coordination geometries as well as a variety of chemical valence states. During the past years, significant progress has been made in the syntheses of polyoxovanadates by incorporating group 15 elements (As^III^/Sb^III^, vanadoarsenates [25,26,27,28,29,30,31,32,33,34,35,36,37,38] and vanadoantimonates [39,40,41,42,43,44,45,46,47,48,49,50,51,52]) into the well-known {V_18_O_42_} shell. In 2015, Monakhov, Bensch and Kögerler published a milestone review on derivatives of polyoxovanadates [53], in which the syntheses and structures of vanadoarsenates, vanadoantimonates and vanadogermanates were systematically reviewed. In addition, we have focused on preparations of vanadoarsenates [25,26,27,28,29,30], vanadoantimonates [40,41] and secondary transition metal substituted As-V-O clusters [54] for years. Here, we further extended our interest in Ge-V-O [55,56,57,58,59,60,61,62,63,64,65] and secondary transition metal substituted Ge-V-O clusters [6,66] based on two considerations. Firstly, the {As^III^_2_O_5_} of the As-V-O cluster is not favorable for forming extended structures because the arsenic center did not have terminal oxygens, which can further interact with other bridging metal centers, whereas the {Ge^IV^_2_O_7_} of the Ge-V-O cluster has two additional terminal oxygens, which can provide opportunities for forming extended structures via metal-oxygen covalent and dative bonds. Secondly, like the As-V-O cluster, some vanadiums of the Ge-V-O cluster can also be substituted by secondary transition metals to yield new organic–inorganic hybrid clusters [54]. Vanadogermanates can significantly expand the area of polyoxovanadate chemistry due to the introduction of a different functionality compared to the As-containing congeners. In 2003, A. J. Jacobson [57], A. Clearfield [60] and Lin [56] respectively reported the preparations of a series of Ge-V-O compounds, and then W. Bensch reported several Ge-V-O compounds in 2006, 2010 and 2013 [59,61,64]. In 2010 and 2014, Yang reported the syntheses of several secondary transition metal-substituted Ge-V-O clusters [6,66]. However, compared with vanadoarsenates, the number of vanadogermanates is still far too small, and especially the secondary transition metal substituted Ge-V-O clusters. It is still a great challenge for chemists to synthesize new vanadogermanates.

We found that all the previously reported Ge-V-O compounds were totally based on aliphatic organic ligands [6,39,40,41,42,43,44,45,46,47,48,49,50,51,52,66], while no Ge-V-O compounds constructed out of aromatic organic ligands were reported. The reason only aliphatic-ligand involving Ge-V clusters were reported can be listed as below: (1) GeO_2_ is inert in neutral and acidic aqueous solutions; (2) the aqueous solution of the aromatic nitrogen-containing organic ligands is neutral. It is not favorable for the aggregation of Ge-V clusters. Therefore, it is very difficult to prepare aromatic-ligand-containing Ge-V clusters. The first Ge-V clusters were reported in 2003 [60], and no aromatic-ligand-containing Ge-V clusters have been prepared. On the other hand, the introduction of aromatic organic ligands can not only can enrich the structures of this kind of compound but can also ameliorate their polar, electricity, acid and redox properties [67,68,69,70]. The introduction of aromatic organic ligands may thereby lead to compounds with more interesting structures, topologies and properties (It is well known that the robustness of almost all MOFs is derived from the aromatic organic ligands [71]). An example: recently, S. K. Das reported an aromatic-ligand-containing polyoxometalate that can be used as an efficient electrocatalyst for water oxidation [72], but the aliphatic analog did not exhibit such an excellent electrocatalytic property. Based on aforementioned points, we then chose phen as the aromatic organic ligand to prepare Ge-V-O compounds. Fortunately, we successfully synthesized [Cd(phen)(en)]_2_[Cd_2_(phen)_2_V_12_O_48_Ge_8_(OH)_8_(H_2_O)]∙12.5H_2_O (**1**), [Cd(DETA)]_2_[Cd(DETA)_2_]_0.5_[Cd_2_(phen)_2_V_12_O_41_Ge_8_(OH)_7_(0.5H_2_O)]∙7.5H_2_O (**2**) and [Cd(en)_3_]{[Cd(η_2_-en)_2_]_3_[Cd(η_2_-en)(η_2_-en)(η_2_-μ_2_-en)Cd][Ge_6_V_15_O_48_(H_2_O)]}∙5.5H_2_O (**3**), of which compounds **1** and **2** are the first Ge-V-O compounds based on aromatic organic ligands. Compound **1** is the first dimer of Ge-V-O compound, of which Ge-V clusters are linked by a double bridge of [Cd(phen)(en)]^2+^. Compound **2** exhibits a novel 1-D chain structure of which Ge-V-O clusters are fused by [Cd_2_(DETA)_2_]^4+^ TMCs. Compound **3** is a novel 3-D structure which is constructed out of [Ge_6_V_15_O_48_(H_2_O)]^12−^ clusters and five different types of TMCs. We also synthesized [Zn_2_(enMe)_3_][Zn(enMe)]_2_[Zn(enMe)_2_(H_2_O)]_2_[Ge_6_V_15_O_48_(H_2_O)]∙3H_2_O (**4**) [6] and [Cd(en)_2_]_2_{H_8_[Cd(en)]_2_Ge_8_V_12_O_48_(H_2_O)}∙6H_2_O (**5**), which have been reported previously [54]. In addition, the catalytic properties of these five compounds have been investigated. 

## 2. Experimental Section

### 2.1. Chemicals and Data Analysis

All the chemicals used were of reagent grade without further purification. C, H, N elemental analyses were carried out on a Perkin-Elmer 2400 CHN elemental analyser (Shanghai, China). Infrared spectra were recorded as KBr pellets on a Perkin-Elmer SPECTRUM ONE FTIR spectrophotometer. UV-vis spectra were recorded on a Shimadzu UV-3100 spectrophotometer. Powder XRD patterns were obtained with a Scintag X1 powder diffractometer system using Cu Kα radiation with a variable divergent slit and a solid-state detector. Electron spin resonance (ESR) spectra were performed on a JEOL JES-FA200 spectrometer(Guangzhou, China) operating in the X-band mode. The g value was calculated by comparison with the spectrum of 1,1-diphenyl-2-picrylhydrazyl (DPPH), whereas the spin concentrations were determined by comparing the recorded spectra with that of an Mn marker and DPPH, using the built-in software of the spectrometer.

### 2.2. Syntheses of Compounds Based on Ge-V-O Clusters

#### 2.2.1. [Cd(phen)(en)]_2_[Cd_2_(phen)_2_V_12_O_40_Ge_8_(OH)_8_(H_2_O)]∙12.5H_2_O (**1**)

V_2_O_5_ (0.061 g, 0.33 mmol), GeO_2_ (0.069 g, 0.67 mmol) and TMAH (TMAH = tetramethyl-ammonium hydroxide) (0.10 mL) were added to H_2_O (3.00 mL) solution with stirring for a half-hour. Then, CdCl_2_ (0.061 g, 0.33 mmol), phen (0.066 g, 0.33 mmol) and 2,2′-bpy (2,2′-bpy = 2,2′-bipyridine, 0.052 g, 0.33 mmol) were added, the resulting suspension was further stirred for 4 h, the pH of the mixture was 5.0. Finally, the pH of the mixture was adjusted to 9.5 with en, which was stirred for another 0.5 h and then was sealed in a Teflon-lined stainless bomb and heated at 170 °C for 5 days. Brown rectangle crystals were collected by filtration and washed with water (Yield: 0.149 g, 51.60% based on GeO_2_). Compound **1** can also be prepared by adjusting the pH to 10.0. When the pH of the mixture was adjusted to 9.5, more crystals were obtained, and the crystal quality was better. Anal. Calcd for C_52_H_83_Cd_4_Ge_8_N_12_O_61.5_V_12_: C, 17.83; H, 2.39; N, 4.80%. Found: C, 17.71; H, 2.28; N, 4.83%. 

#### 2.2.2. [Cd(DETA)]_2_[Cd(DETA)_2_]_0.5_[Cd_2_(phen)_2_V_12_O_41_Ge_8_(OH)_7_(0.5H_2_O)]∙7.5H_2_O (**2**)

V_2_O_5_ (0.067 g, 0.36 mmol), GeO_2_ (0.069 g, 0.67 mmol) and TMAH (0.10 mL) were added to the H_2_O (3.00 mL) solution with stirring for a half-hour. Then, CdCl_2_ (0.183 g, 1 mmol) and phen (0.066 g, 0.33 mmol) were added, and the resulting suspension was further stirred for 4 h; the pH of the mixture was 5.0. Finally, the pH of the mixture was adjusted to 9.5 with DETA solution, which was stirred for 0.5 h, and then was sealed in a Teflon-lined stainless bomb and heated at 170 °C for 5 days. Black needle crystals were collected by filtration and washed with water (Yield: 0.086 g, 31.30% based on GeO_2_). Anal. Calcd for C_36_H_78_Cd_4.5_Ge_8_N_13_O_56_V_12_: C, 13.15; H, 2.39; N, 5.54%. Found: C, 12.91; H, 2.40; N, 5.33%.

#### 2.2.3. [Cd(en)_3_]{[Cd(η_2_-en)_2_]_3_[Cd(η_2_-en)(η_1_-en)(η_2_-en)Cd][Ge_6_V_15_O_48_(H_2_O)]}∙5.5H_2_O (**3**)

GeO_2_ (0.104 g, 1.00 mmol), NH_4_VO_3_ (0.2323 g, 2.00 mmol) and CdCl_2_ (0.1831 g, 1.00 mmol) were added to a 25% aqueous solution of en (6.00 mL). The resulting suspension was further stirred for 12 h, then 2, 2′-bpy (0.156 g, 1.0 mmol) were added, the final mixture (pH 9.7–10) was moved to a 35 mL Teflon-lined autoclave, sealed and kept at 170 °C for 5 days and then it was cooled to ambient temperature. Black square crystals were collected by filtration and washed with water (Yield: 0.413 g, 71.20% based on GeO_2_). Anal. Calcd for C_24_H_109_Cd_6_Ge_6_N_24_O_54.5_V_15_: C, 8.28; H, 3.16; N, 9.66%. Found: C, 8.19; H, 3.00; N, 9.63%.

### 2.3. X-ray Crystallography

The crystal data for compound **1** were measured on a Bruker Apex II diffractometer with graphite monochromated Mo Kα (λ = 0.71073 Ǻ) radiation. The data for compounds **2** were measured on a Rigaku R-AXIS RAPID diffractometer with graphite monochromated Mo Kα (λ = 0.71073 Ǻ) radiation, while the data for compound **3** were measured on an Agilent Technology SuperNova Eos Dual system with a Mo Kα (λ = 0.71073 Ǻ) microfocus source and focusing multilayer mirror optics. None of the crystals showed evidence of crystal decay during the data collections. Refinements were carried out with SHELXS-2014/7 [73] and SHELXL-2014/7 [73] using Olex 2.0 interface via the full matrix least-squares on F2 method. In the final refinements, all atoms were refined anisotropically in compounds **1**–**3**. The hydrogen atoms of en, phen, DETA and enMe in the three compounds were placed in calculated positions and included in the structure factor calculations but not refined. In these heavy-atom structures with reflection data from poor-quality crystals it was not possible to see clear electron-density peaks in difference maps which would correspond with acceptable locations for the various H atoms bonded to water oxygen atoms. The refinements were then completed with no allowance for these water H atoms in the models; the CCDC number: 1,525,920 for 1, 2,024,572 for 2 and 1,525,922 for 3. The reflection intensity data for compounds **4** and **5** were also measured on a Rigaku R-AXIS RAPID diffractometer with graphite monochromated Mo Kα (λ = 0.71073 Ǻ) radiation, and the results show that the two compounds have already been reported previously [6]. A summary of the crystallographic data and structure refinements for compounds **1**–**3** is given in Table 1.

## 3. Results and Discussion

### 3.1. Synthesis Description

Compounds **1** and **2** are all based on Cd_2_Ge_8_V_12_ and compounds **3** and **4** are based on Ge_6_V_15_. The alkalinity (pH > 9) and the stirring time of the reaction mixture are important for the formation of Ge_6_V_15_ in compounds **3** and **4**. We have a relatively clear grasp of the synthetic conditions of the two different clusters. The molar ratio of V_2_O_5_ to GeO_2_ for compounds **1** and **2** is about 1:2, and the molar ratios of NH_4_VO_3_ to GeO_2_ for compounds **3** and **4** are 2:1. The addition of 2, 2′-bpy is important for the preparations of compounds **2** and **3**. Though it is absent in the products, 2, 2′-bpy is required for the syntheses of compounds **2** and **3**. It should be noted that such a phenomenon is not unusual in hydrothermal preparations [74].

### 3.2. Description of Crystal Structures

#### 3.2.1. [Cd(phen)(en)]_2_[Cd_2_(phen)_2_V_12_O_40_Ge_8_(OH)_8_(H_2_O)]∙12.5H_2_O (**1**)

The asymmetric unit of **1** consists of a di-Cd-substituted Ge-V-O cluster [H_8_Cd_2_(phen)_2_Ge_8_V_12_O_48_(H_2_O)]^4−^ (Cd_2_Ge_8_V_12_), two [Cd(phen)(en)]^2+^ and 12.5 water molecules. As shown in Figure 1, an unusual feature of **1** is that two [Cd(phen)]^2+^ take the place of the two VO^2+^ fragments located at the two opposite positions of {Ge_8_V_14_O_50_} [59], forming Cd_2_Ge_8_V_12_. The two substituted cadmiums each is coordinated by four oxygens from two {Ge_2_O_7_} units with Cd-O distances of 2.290(5)–2.366(5) Å, and two nitrogens from a phen ligand with Cd-N distances of 2.353(6)–2.427(7) Å. That is to say, two phen ligands were decorated onto the surface of Cd_2_Ge_8_V_12_. The two phen located at the two sides of Cd_2_Ge_8_V_12_ are not parallel to each other. There is a dihedral angle of 36.116° between the two phenanthroline-planes. All the bond distances in **1** are comparable to those of previously reported compounds [6,55,56,57,58,59,60,61,62,63,64,65,66]. Bond valence sum (BVS) calculations for Ge and V indicate that both Ge and V exist in the +4 oxidation-state (Appendix A). BVS calculations were also conducted for the cadmium and oxygen atoms in compound **1** to determine the locations of the hydrogen atoms in compound **1** (see Appendix A and discussions in “BVS calculations to determine the locations of hydrogen atoms for compounds **1**–**3**”) [75]. 

Except for [Cd(phen)]^2+^, there are two [Cd(phen)(en)]^2+^. It should be noted that the two [Cd(phen)(en)]^2+^ are different from each other. Cadmium of [Cd(4)(phen)(en)]^2+^ of the two is bonded to four nitrogens from a phen and an en with Cd-N distances of 2.250(8)–2.320(7) Å, a terminal oxygen from Cd_2_Ge_8_V_12_ with the Cd-O distance of 2.385(5) Å and a water molecule with the Cd-O distance of 2.583(9) Å, exhibiting a cis-octahedral geometry. Therefore, the cluster acts as a monodentate inorganic ligand coordinating with Cd(4), forming a cluster supported transition metal complex (TMC). Cadmium of [Cd(3)(phen)(en)]^2+^ of the two receives contributions from four nitrogens from a phen and an en with Cd-N distances of 2.290(7)–2.367(7) Å and two terminal oxygens from two Cd_2_Ge_8_V_12_ with Cd-O distances of 2.284(5)–2.443(5) Å. It should be noted that the two terminal oxygens involving Cd-O bonds are distinct: one is from a {Ge_2_O_7_}, but the other comes from a {VO_4_}. Thus, Cd(3) TMC acts as a bridge linking two Cd_2_Ge_8_V_12_ to construct a novel cluster dimer. It should be noted that there are two Cd(3) TMCs acting as a double bridge linking two Cd_2_Ge_8_V_12_. The dimer further supports two Cd(4) TMCs at the two sides of the dimer. That is to say, Cd(3) TMCs act as bridges joining Cd_2_Ge_8_V_12_, but Cd(4) TMCs terminate the connection of the clusters by the terminating water molecule. 

Distances between the central water molecule of the Cd_2_Ge_8_V_12_ and Cd(3) and Cd(4) are 7.886–7.888 Å, and the angle of Cd(3)-O1w-Cd(4) is 109.754(1)°. 

The dimer of clusters was reported by our group in 2002 [76] and very recently [77]; the first one was based on the Mo_8_V_6_ cluster, and the second one was based on the V_15_O_36_ cluster. However, compound **1** here is the most complex one of the three, which is the first example of dimer of substituted clusters. The other two reported compounds are both based on traditional clusters but not the substituted one.

#### 3.2.2. Cd(DETA)]_2_[Cd(DETA)_2_]_0.5_[Cd_2_(phen)_2_V_12_O_41_Ge_8_(OH)_7_(0.5H_2_O)]∙7.5H_2_O (**2**)

The building block [H_7_Cd_2_(phen)_2_Ge_8_V_12_O_48_(0.5H_2_O)]^5−^ (Cd_2_Ge_8_V_12_) of **2** is almost identical to that of **1**, which is also a cadmium di-substituted Ge-V-O cluster; each substituted cadmium is also coordinated by a phen ligand. The main difference between the building blocks of compounds **2** and **1** is the number of the attached hydrogen atoms. There are only slight differences between the bond lengths and angles in compounds **2** and **1**. Bond valence sum calculations for Ge and V also indicate that Ge and V are in the +4 oxidation-state (Appendix A).

Except for [Cd(phen)]^2+^ TMCs, there are two different TMCs which are [Cd(DETA)_2_]^2+^ and [Cd(DETA)]^2+^ (Figure 2). The two TMCs are thoroughly different from those in **1**. Cadmium of [Cd(DETA)_2_]^2+^ is bound to six nitrogens from two DETA ligands and a terminal oxygen from Cd_2_Ge_8_V_12_ with Cd-O and Cd-N distances of 2.46(1) and 2.38(3)–2.52(3) Å. [Cd(DETA)_2_]^2+^, performing a similar role as Cd(4) TMC in compound **1**, serves as a TMC supported by Cd_2_Ge_8_V_12_. Cd of [Cd(DETA)]^2+^ is bonded to three nitrogens from a DETA with Cd-N distances of 2.26(1)–2.40(1) Å and two terminal oxygens from two {Ge_2_O_7_} from two adjoining Cd_2_Ge_8_V_12_ with Cd-O distances of 2.234(8)–2.240(8) Å, exhibiting a five-coordinated trigonal bipyramidal geometry. Cadmium of [Cd(DETA)]^2+^ serves as a bridge connecting the two Cd_2_Ge_8_V_12_. It should be noted that the two terminal oxygens was shared by the two [Cd(DETA)]^2+^, meaning that two terminal oxygens simultaneously connect two [Cd(DETA)]^2+^ to form a novel dimer [Cd_2_(DETA)_2_O_2_]. The role of [Cd_2_(DETA)_2_O_2_] in compound **2** is only partly similar to that of Cd(3) TMC in compound **1**. Two Cd(3) TMCs serving as a double bridge links two Cd_2_Ge_8_V_12_ to form a dimer in compound **1**, but [Cd_2_(DETA)_2_O_2_] in compound **2** acting as a single bridge connects two Cd_2_Ge_8_V_12_, and for its two components [Cd(DETA)]^2+^, is also joined by the two terminal oxygens to form a single building unit. Most importantly, [Cd_2_(DETA)_2_O_2_] in compound **2** connects Cd_2_Ge_8_V_12_ to form a novel 1-D extended chain structure. It should be noted that the neighboring Cd_2_Ge_8_V_12_ in the extended chain are oriented up and down, as shown in Figure 2. To our knowledge, compound **2** is the first extended structure based on a metal-substituted Ge-V-O cluster of aromatic organic ligands. Yang et. al. also reported a 1-D chain structure formed by similar substituted Ge-V-O clusters and coordination fragments [54]. However, Yang’s cluster is based on aliphatic organic ligands but not aromatic organic ones. Secondly, Yang’s coordination fragment is formed by en ligands rather than DETA ligands. Finally, the 1-D chain of Yang’s compound is sinusoidal, but the one here is linear. 

#### 3.2.3. [Cd(en)_3_]{[Cd(η_2_-en)_2_]_3_[Cd(η_2_-en)(η_2_-μ_2_-en)(η_2_-en)Cd][Ge_6_V_15_O_48_(H_2_O)]}∙5.5H_2_O (**3**)

The asymmetric unit of compound **3** is composed of [Ge_6_V_15_O_48_(H_2_O)]^12−^ (Ge_6_V_15_), [Cd(η_2_-en)_2_]^2+^, [Cd(η_2_-en)(η_1_-en)]^2+^, [Cd(η_2_-en)_3_]^2+^ and 5.5 water molecules. The framework of the cluster in compound **3** is similar to those of {As_6_V_15_O_42_} [16,17,18] and {Sb_6_V_15_O_42_} [19,20,21,22,23,24], with {Ge_2_O_7_} displacing {As_2_O_5_} and {Sb_2_O_5_} in {As_6_V_15_O_42_} and {Sb_6_V_15_O_42_}. Although the oxo-cluster in compound **3** is thoroughly different from those in compounds **1** and **2**, the bond lengths and angles in compound **3** are comparable to those in compounds **1** and **2**. Bond valence sum calculations for Ge and V reveal that oxidation states of both Ge and V are +4 (Appendix A).

It should be noted that [Cd(η_2_-en)_2_]^2+^ of the five has two different configurations (Figure 3a). Cd(3) of [Cd(η_2_-en)_2_]^2+^, which exhibits a trans-octahedral geometry, is bonded to four nitrogens from two en and two oxygens from two Ge_6_V_15_ with Cd-N and Cd-O distances in the range of 2.26(1)–2.31(2) Å and 2.229(9)–2.242(9) Å. Therefore, the trans-octahedral Cd(3) TMC joins two Ge_6_V_15_. Cd(5) of [Cd(η_2_-en)_2_]^2+^ has a cis-octahedral geometry, which is coordinated by four nitrogens from two en with Cd-N distances of 2.33(2)–2.44(2) Å and two terminal oxygens in two cis-positions from two Ge_6_V_15_ with Cd-O distances of 2.228(7)–2.337(7) Å. Thus, the cis-octahedral Cd(5) TMC also connects two Ge_6_V_15_. Although Cd(3) and Cd(5) TMCs show different configurations, both their terminal oxygen atoms come from {Ge_2_O_7_} units of Ge_6_V_15_.

There are also two different [Cd(η_2_-en)(η_2_-μ_2_-en)]^2+^ TMCs in compound **3**. [Cd(6)(η_2_-en)(η_2_-μ_2_-en)]^2+^ presents a six-coordinated octahedral geometry with two nitrogens from a η_2_-en, one nitrogen from a η_2_-μ_2_-en and three oxygens from two Ge_6_V_15_ with Cd-N and Cd-O distances of 2.30(1)–2.37(1) Å and 2.235(8)–2.610(8) Å (the first one oxygen is from one Ge_6_V_15_ and the remaining two oxygens are from the other Ge_6_V_15_). Cd(6) also serves as a bridge linking two Ge_6_V_15_. It should be noted that two oxygens of Cd(6) octahedron from two Ge_6_V_15_ are shared by Cd(5) octahedron. [Cd(4)(η_2_-en)(η_2_-μ_2_-en)]^2+^ is only five-coordinated by two nitrogens from a η_2_-en, one nitrogen from a η_2_-μ_2_-en, and two oxygen atoms from two Ge_6_V_15_ with Cd-N and Cd-O distances of 2.29(1)–2.38(1) Å and 2.228(8)–2.238(8) Å, exhibiting a square pyramidal geometry. Cd(4) and Cd(6) are linked by η_2_-μ_2_-en to form a dumbbell-like dimer [Cd(η_2_-en)(η_2_-μ_2_-en)(η_2_-en)Cd]^4+^. All five TMCs serve as bridges linking their neighboring clusters to form a novel 3-D framework structure. It should be noted that two terminal oxygens of Cd(1) octahedron are also shared by Cd(4) pyramid. 

With the exception of the four different TMCs, there is a dissociated TMC [Cd(η_2_-en)_3_]^2+^. Cd(2) of [Cd(η_2_-en)_3_]^2+^ is chelated by three en with Cd-N distances in the range of 2.36(1)–2.40(1)Å. [Cd(μ_2_-en)_3_]^2+^ did not interact with any Ge_6_V_15_, which only serves as the space-filling agent and counterion. 

In conclusion, there are five types of TMCs in compound **3**. To the best of our knowledge, compound **3** contains the largest number of TMC types.

The TMCs and the Ge_6_V_15_ clusters are fused to form a novel 3-D framework structure via Cd-O covalent interactions, and the framework exhibits channels running along the [101], [110] and [011] directions. As shown in Figure 3, the framework exhibits gold ingot-shaped pores along the [101] direction. It should be noted that there are two kinds of such pores with different orientations. The framework exhibits dumbbell-shaped pores along the [110] direction; there are also two kinds of such pores with different orientations. The framework exhibits cross-shaped pores along the [011] direction; the pores here exhibit two orientations as well. The three kinds of channels intersect one another. Yang et al. also reported a 3-D structure formed by similar Ge-V-O clusters and coordination fragments [6]. However, there are several significant differences between our compound and Yang’s compound. Firstly, and most importantly, the Ge-V-O cluster of Yang’s compound is Ge_4_V_16_, but the corresponding cluster of our compound is Ge_6_V_15_. Secondly, Yang’s compound is based on diethylenetriamine ligands but not en in our compound. Finally, Yang’s compound did not exhibit various channels that were found in our compound.

### 3.3. BVS Calculations to Determine the Locations of Hydrogen Atoms of Compounds **1**–**3**

Single crystal X-ray diffraction cannot exactly determine the positions of the hydrogen atoms from the Fourier maps. For further verifying the correctness of the formula of the three compounds, BVS calculations [75] were carried out to determine the positions of the hydrogen atoms for all the three compounds. As for compound **1**, the oxygens can be classified into eight groups: (1) seven Ge-O_t_ terminal oxygens; (2) one Ge-O_t_-Cd μ_2_-oxygen; (3) eleven V-O_t_ terminal oxygens; (4) one V-O_t_-Cd μ_2_-oxygen; (5) eight μ_3_-oxygens located between two vanadiums and one germanium; (6) eight μ_3_-oxygens located between three vanadiums; (7) eight μ_3_-oxygens between a vanadium, cadmium and germanium; and (8) four μ_2_-oxygens between two germaniums. All the atoms of the eight groups except groups (1) and (2) can be assigned to the −2 valence state, with BVS calculation results in the range of 1.56–2.16. With respect to the group (1) oxygens, all seven oxygens exist in the -1 valence state, with BVS results ranging from 1.01–1.04, indicating that all seven terminal Ge-O_t_ oxygens are mono-protonated. The BVS value of the group (2) oxygen is 1.38, meaning that although this oxygen is coordinated by both one cadmium and one germanium, it exists in the −1 valence state. Therefore, the cluster in compound **1** is attached by eight hydrogens, and all eight hydrogens are attached on the eight Ge-O terminal oxygens.

As for compound **2**, the oxygens can also be divided into eight groups. Seven of the eight groups are similar to the corresponding groups in compound 1. Only the eighth one is not found in compound **1**: it is a μ_3_-oxygen between two cadmiums and a germanium. This μ_3_-oxygen is a terminal oxygen from a {Ge_2_O_7_} simultaneously interacting with two cadmiums and one germanium. Therefore, its valence state is not −1 but −2, with the BVS result of 1.85. In conclusion, only six of the eight terminal Ge-O_t_ oxygens are mono-protonated. Thus, there is still one hydrogen atom whose position cannot be determined. We think this hydrogen should be disorderedly distributed on the surface of the cluster. 

There are also seven groups of oxygens in compound **3**. However, only five of the seven have corresponding groups in compound **1**. The five groups are: (1) V-O**_t_** terminal oxygens; (2) μ**_3_**-oxygens between two vanadiums and one germanium; (3) μ**_3_**-oxygens between three vanadiums; (4) μ**_2_**-oxygens between two germaniums; and (5) μ**_2_**-oxygen between one terminal vanadium and one cadmium. The remaining two groups are: (6) μ**_3_**-oxygen between two cadmiums and one germanium, which has the corresponding group in compound **2**; and (7) μ**_2_**-oxygen between one cadmium and one germanium, which is only observed in compound **3**. Compound **3** did not contain Ge-O**_t_** terminal oxygens, and all the Ge-O**_t_** terminal oxygens simultaneously interact with one or two cadmiums and finally form the group (6) and (7) oxygens. For the contributions of the cadmiums of group (6) and (7) oxygens, the BVS values of these oxygens are in the range of 1.56–2.01, indicating that there are no hydrogens attached on the cluster in compound **3**.

### 3.4. IR Spectrophotometry

The IR spectra of compounds **1**–**4** were recorded in the regions between 4000 and 200 cm^−1^ (Appendix A). The strong peak at 984 cm^−1^ of compound **1** can be attributed to the stretching vibration of V=O. The patterns of the bands in the region characteristic of ν(V=O_t_) indicate the presence of V^IV^ sites: clusters which contain exclusively V^IV^ generally possess ν(V=O_t_) bands in the range of 970–1000 cm^−1^, while bands in the region 940–960 cm^−1^ are characteristic of V^V^. The observation of a strong absorbance in the 970–1000 cm^−1^ region provides a useful diagnostic for the presence of V^4+^ centers [78]. The strong peaks at 793 and 821 cm^−1^ of compound **1** may be due to asymmetric Ge-O stretching vibrations of {GeO_4_}. The infrared spectrum of compound **2** is very similar to that of compound **1**. It also shows characteristic peaks at 983 cm^−1^ and 788 cm^−1^, which should be ascribed to V=O_t_ and Ge-O vibrations in compound **2**. 

Compounds **3** and **4** are based on Ge_6_V_15_, which is different from that of compounds **1** and **2**. However, it should be noted that Ge_6_V_15_ is also formed by {GeO_4_} and {V^IV^O_5_}; thus, the IR spectra of compounds **3** and **4** are very similar to those of compounds **1** and **2**. The IR spectra of compounds **3** and **4** present characteristic peaks at 979, 801 cm^−1^ and 982, 800 cm^−1^, respectively, which correspond to V=O_t_ and Ge-O vibrations in compounds **3** and **4**. The main difference between the IR spectra of compounds **1** and **2** and **3** and **4** is that the bands at 667 and 660 cm^−1^ of compounds **1** and **2** are weak, but the corresponding bands at 691 and 692 cm^−1^ for compounds **3** and **4** are much stronger. Bands of 667–692 cm^−1^ can be ascribed to V-O-V vibrations.

### 3.5. XRD Powder Diffractometer

The powder X-ray diffraction patterns for compounds **1**–**4** are all in good agreement with the ones simulated based on the data of the single-crystal structures, indicating the purity of the as-synthesized products (Appendix A). The differences in the reflection intensity are probably due to preferred orientations in the powder samples of compounds **1**–**4**. 

### 3.6. UV-Vis Spectrophotometry

The UV-vis spectra of compounds **1**–**4**, in the range of 250–600 nm, are presented in Appendix A. The UV-Vis spectrum of compound **1** displays an intense absorption sharp peak centered at about 266 nm, a shoulder peak at 294 nm and a peak tailing to the longer wavelength side (to about 450 nm), which can be assigned to O→V charge transfer, n→π* transitions of phen ligands and d→d transitions of complexes in compound **1**. The UV-Vis spectrum of compounds **2** displays an intense absorption peak at about 265 nm assigned to the O→V charge transfer in the polyoxoanion structure of compound **2**. The peak corresponding to the n→π* transitions of phen ligands was overlapped by the O→V charge transfer and cannot be separated.

The UV spectra of compounds **3** and **4** are similar to each other, but are different from those of compounds **1** and **2**, which exhibit absorption peaks at about 254 and 255 nm due to the O→V charge transfer in compounds **3** and **4**. The difference in the UV-Vis spectra between compounds **3**–**4** and compounds **1**–**2** may be due to the difference in their clusters.

### 3.7. ESR Spectrophotometry

The ESR spectra of compounds **1**–**4** were studied at room temperature (Appendix A). The ESR spectra of compounds **1**–**4** are very similar to one another, which show Lorentzian shapes accompanied by signals at g = 1.968, 1.968, 1.912 and 1.941, respectively, indicating that the vanadium atoms in compounds **1**–**4** are in a +4 oxidation-state. The ESR spectra further confirm the results of the bond valence sum calculations for compounds **1**–**4**.

### 3.8. Catalytic Activity

Epoxidation is an important industrial reaction, and epoxides are key intermediates in the manufacture of a wide variety of valuable products [79,80,81]. The epoxidation of styrene to styrene oxide with aqueous tertbutyl hydroperoxide (TBHP) using compound **1**, **2**, **3**, **4** or **5** as the catalyst was carried out in a batch reactor. In a typical run, the catalyst (compound **1** (2 mg, 0.57 μmol), compound **2** (2 mg, 0.60 μmol), compound **3** (2 mg, 0.58 μmol), compound **4** (2 mg, 0.62 μmol), compound **5** (2 mg, 0.70 μmol), 0.114 mL (1 mmol) of styrene and 2 mL of CH_3_CN were added to a 10-mL two-neck flask equipped with a stirrer and a reflux condenser. The mixture was heated to 80 °C and then 2 mmol of TBHP was injected into the solution to start the reaction. The liquid organic products were quantified using a gas chromatograph (Shimadzu, GC-8A, Beijing, China) equipped with a flame detector and an HP-5 capillary column and identified by comparison with authentic samples and GC-MS coupling. In a blank experiment carried out in the absence of catalyst, no products were observed. Also, the styrene epoxidation reactions in the presence of GeO_2_ (2 mg, 19.1 μmol) and V_2_O_5_ (2 mg, 11.0 μmol) were carried out respectively, and the activities are 24.8% and 71.2%, respectively, after 8 h.

Table 2 shows the catalytic reaction results of TBHP oxidation of styrene over various catalysts. As expected, all the catalysts are active for the oxidation of styrene. Compound **1** as a catalyst shows a performance with 50.1% conversion and 62.8% selectivity to styrene oxide after 8 h. Compound **2** shows the highest activity among the five with 96.3% conversion and 71.6% selectivity to styrene oxide. Compound **3** shows a catalytic performance with 81.4% conversion and 63.0% selectivity. The performance of compound 4 is similar to that of compound **3** with 84.1% conversion and 55.5% selectivity. The activity and selectivity of compound **5** are 41.7% and 67.1%, respectively. Compounds **3** and **4** are based on Ge_6_V_15_, group 12 metals (Cd and Zn) and similar organic ligands (en and enMe), and both exhibit extended framework structures (3-D and 2-D). Therefore, the catalytic activities of the two are similar. The structures of compounds **2** and **5** are more similar to each other. Compounds **2** and **5** are based on similar Cd_2_Ge_8_V_12_ clusters and similar cadmium complexes, and both exhibit similar 1-D extended structures. The significant difference between compounds **2** and **5** is that compound **2** contains aromatic organic ligands but compound **5** dose not; however, the catalytic activities of the two are thoroughly different from each other. To further understand the catalytic mechanism, we still need not only more Ge-V-O crystals but also more catalytic experimental results of the synthesized crystals. Although there have been no investigations on Ge-V-O metal-oxo-clusters as catalysts, there are some similar catalysis studies using catalysts formed by other POMs. The comparisons of the catalytic oxidation of styrene for compounds **1**–**5** and other reported POMs have been summarized in Appendix A.

The recyclability and reusability of compound **3**, including the conversion and catalyst recovery in three cycles, were studied (Table 3). The same experimental conditions were used. Generally, when using soluble heteropolyacid (e.g., H_3_[PW_12_O_40_]) as the catalyst, the used catalyst was recovered by precipitation and ion exchange [82]. In comparison, it was easy to separate (centrifugation) and recycle compound **3**. The process of recovery possibly resulted in the loss of approximately 40 wt.% after each cycle. The conversion dropped from 81.4% to 44.0% after three cycles. 

Recovery experiments showed that compound **3** suffered significant activity losses after three cycles. However, the residual catalyst of compound **3** and the as-synthesized crystals used for X-ray analysis can still be considered homogeneous (Appendix A). The FT-IR spectra of compound **3** after the three cycles also remain identical to the one before the reaction (Appendix A).

## 4. Conclusions

The synthesis of Ge-V-O clusters, especially secondary metal substituted Ge-V-O clusters is still a great challenge for chemists. In this manuscript, we synthesized compounds **1** and **2**, which are the first examples formed by Ge-V-O clusters and transition metal complexes of aromatic organic ligands. Compounds **1** and **2** are also the first secondary metal substituted Ge-V-O clusters of aromatic organic ligands. Compound **3** is a novel 3-D framework with interesting channel structure. The catalytic properties of these compounds and two previously reported compounds have been investigated. We plan to apply these compounds in other oxidation catalytic reactions and hope to find applications of them in electrochemistry as well. 

## Figures and Tables

**Figure 1 molecules-27-04424-f001:**
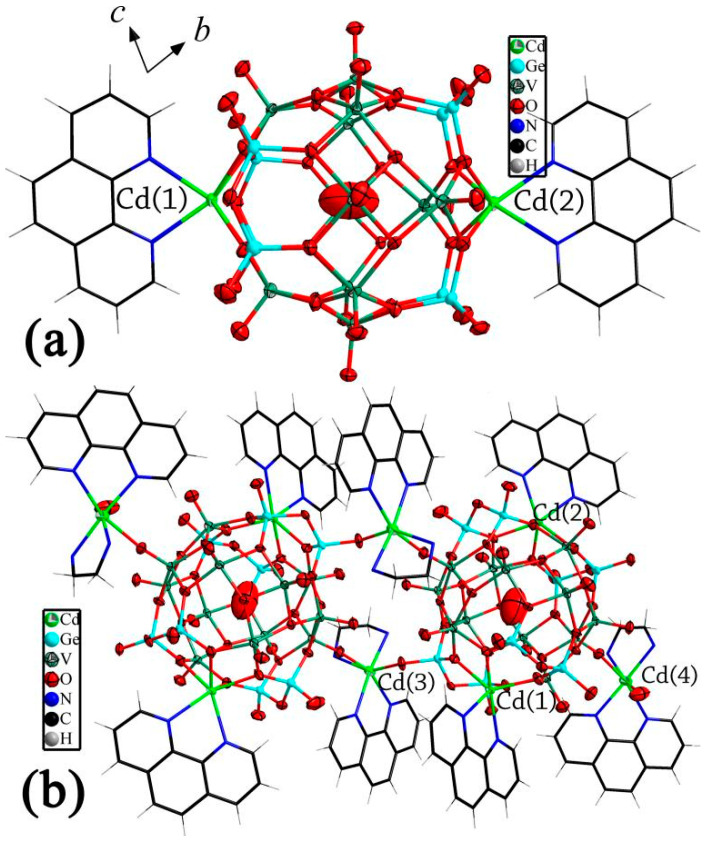
Ball-and-stick and wire representation of the di-Cd-substituted Ge-V-O cluster (**a**) and the dimer in compound **1** (**b**).

**Figure 2 molecules-27-04424-f002:**
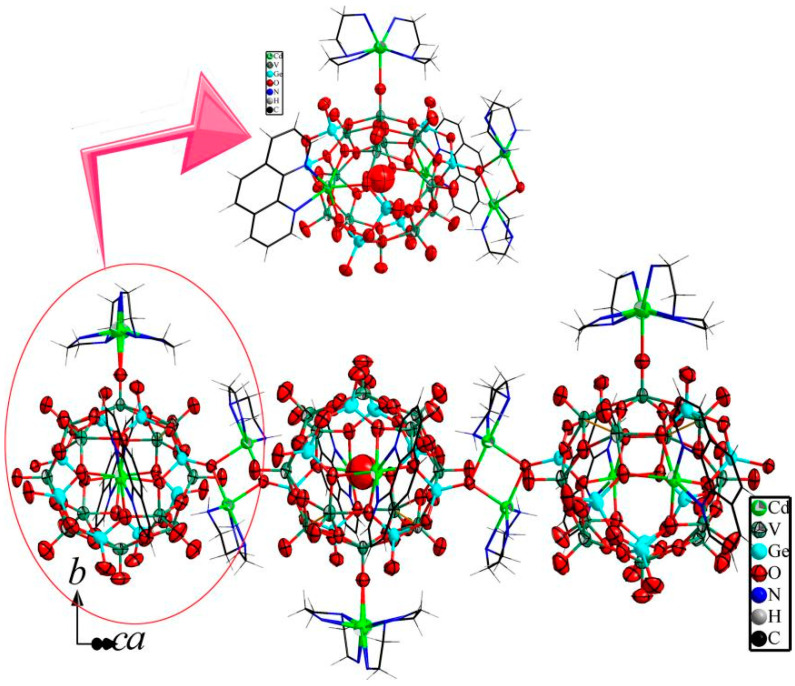
Ball-and-stick and wire representation of the building unit in the 1-D chain structure (**upper**) and the 1-D chain structure formed by Ge-V-O clusters and [Cd_2_(DETA)_2_O_2_] (**lower**).

**Figure 3 molecules-27-04424-f003:**
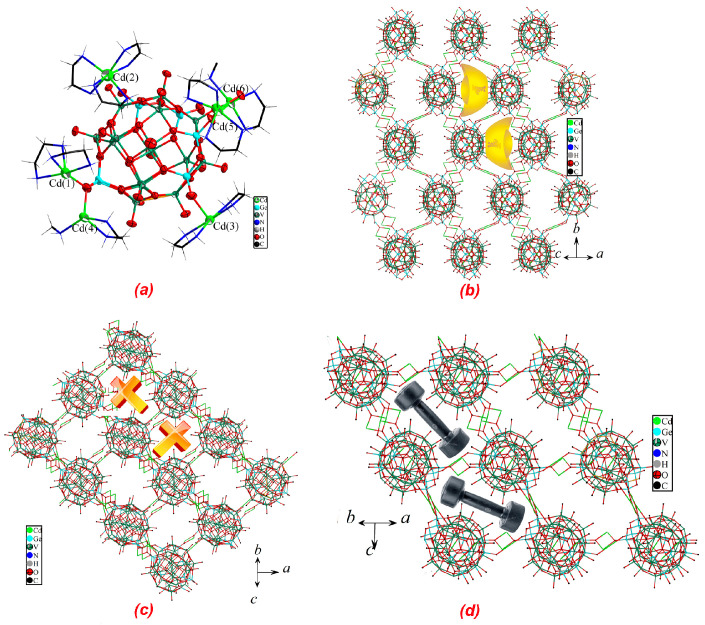
(**a**) Ball-and-stick and wire representation of the [Ge_6_V_15_O_48_]^12−^ cluster and five different types of TMCs in compound 3; (**b**) the framework structure viewed along [101]; (**c**) the framework structure viewed along [011]; (**d**) the framework structure viewed along [110].

**Table 1 molecules-27-04424-t001:** Crystal data and structure refinements for compounds **1**–**3**.

Head 1	Compound 1	Compound 2	Compound 3
Empirical formula	C_52_H_83_Cd_4_Ge_8_N_12_O_61.5_V_12_	C_36_H_78_Cd_4.5_Ge_8_N_13_O_56_V_12_	C_24_H_109_Cd_6_Ge_6_N_24_O_54.5_V_15_
Formula weight	3501.90	3286.91	3480.39
Crystal system	Triclinic	Monoclinic	Monoclinic
space group	P-1	C 2/c	P2_1_/n
*a* (Å)	14.5034(8)	17.193(3)	17.9913(3)
*b* (Å)	16.5920(9)	23.511(5)	23.6117(4)
*c* (Å)	23.0440(13)	26.373(5)	23.9327(4)
*α* (˚)	71.648(4)	90	90
*β* (˚)	84.130(4)	100.15(3)	91.7290(13)
*γ* (˚)	75.454(4)	90	90
Volume (Å^3^)	5093.0(5)	10,494(4)	10,162.1(3)
*Z*	2	4	4
*D_C_* (Mg∙m^−3^)	2.284	2.080	2.275
*μ* (mm^−1^)	4.282	4.242	4.367
*F*(000)	3390	6324	6728
*θ* for data collection	1.375–25.032	3.025–27.466	3.083–29.145
Reflections collected	28,997	45,848	54,419
Reflections unique	17,941	11,814	23,431
*R*(int)	0.1263	0.1080	0.0437
Completeness to θ	99.6	99.1	99.6
parameters	1360	662	1207
GOF on *F*^2^	1.030	1.042	1.027
*R* ^a^ [*I* > 2σ(*I*)]	R1 = 0.0621	R1 = 0.0822	R1 = 0.0780
*R* ^b^ (all data)	ωR2 = 0.1660	ωR2 = 0.2629	ωR2 = 0.2417

^a^ R_1_ = ∑||F_0_| − |F_c_||/∑|F_0_|. ^b^ ωR_2_ = {∑[w (F_0_^2^ − F_c_^2^)^2^]/∑[w(F_0_^2^)^2^]}/^2^.

**Table 2 molecules-27-04424-t002:** Catalytic activity and product distribution.

Catalyst	Styrene Conversion ^a^ (%)	Product Selectivity ^b^ (mol%)		
		S	Bza	Others
GeO_2_	24.8	58.6	39.8	1.7
V_2_O_5_	71.2	67.6	28.6	3.7
Compound **1**	50.1	62.8	34.0	3.2
Compound **2**	96.3	71.6	16.1	12.3
Compound **3**	81.4	63.0	34.8	2.2
Compound **4**	84.1	55.5	39.3	5.1
Compound **5**	41.7	67.1	32.9	0.0

^a^ Reaction conditions: catalyst 2 mg, styrene 0.114 mL (1 mmol), CH_3_CN 2 mL, TBHP (2 mmol), temperature 80 °C and time 8 h. ^b^ So: Styrene oxide, Bza: benzaldehyde; Others: including benzoic acid and phenylacetaldehyde.

**Table 3 molecules-27-04424-t003:** Recyclability and reusability of compound **3**.

Compound 3	Styrene Conversion ^a^ (%)	Product Selectivity ^b^ (mol%)		
		S	Bza	Others
1st run	81.4	63.0	34.8	2.2
2nd run	54.3	59.3	37.8	2.9
3rd run	44.0	43.9	53.0	3.1

^a^ Reaction conditions: catalyst 2 mg, styrene 0.114 mL (1 mmol), CH_3_CN 2 mL, TBHP (2 mmol), temperature 80 °C and time 8 h. ^b^ So: Styrene oxide, Bza: benzaldehyde; Others: including benzoic acid and phenylacetaldehyde.

## Data Availability

Date is available form Corresponding author.

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
