# Peer review of "First Organic–Inorganic Hybrid Compounds Formed by Ge-V-O Clusters and Transition Metal Complexes of Aromatic Organic Ligands"

_molecules, 2022, doi:10.3390/molecules27144424_

Round 1
Reviewer 1 Report
This manuscript reports the synthesis and structural characterization of three metal complexes based on Ge-V-O clusters as well as their catalytic properties towards styrene epoxidation. The authors have paid their effort to prepare this manuscript and I may recommend publication of this manuscript in molecules if the following comments are addressed:
1. Since the structural characterization contributes a major part of this manuscript, the cif and checkcif files of the three new complexes should be submitted to the journal as supplementary materials for review.
2. The formulae of complexes 1 and 2 shown in the manuscript and in the supplementary materials (page 4) are different. Since the authors emphasize that the complexes were prepared under strong basic conditions, the formulae with OH- in supplementary materials should be more acceptable. The eight and seven H in the formulae of complexes 1 and 2 in the manuscript that are used to balance the charge should be proton H+, which should be displayed as counter ions [H]8 and [H]7 and should not be included in the parenthesis as shown in the formulae of 1 and 2. Moreover, the authors should confirm the formation of water molecules, which are important in balancing the charges. It is difficult to differ water from OH- crystallographically.
3. Page 4: “Finally, the 1-D chain of Yang’s compound is sinusoidal but the one here is straight.” It is seldom to use “straight” to describe the structural type of a chain. “Linear” may be better.
4. Figure 2 shows two drawings (a) and (b), but the caption does not tell the difference.
5. Section 2.2. "Synthesis Description” can be positioned ahead of 2.1. “Description of Crystal Structures”.
6. Section 3. “Materials and Methods” can be changed as “Experimental Section”. 3.3. IR Spectrophotometry, 3.5. UV-Vis Spectrophotometry, 3.6. ESR Spectrophotometry and 3.6. Catalytic Activity can be moved to section of 2. Results and Discussion.
Reviewer 2 Report
This paper describes three new compounds based on Ge-V-O polyoxometalates, Cd2+ and chelate N-donor ligands. All compounds were analyzed by a complex of physicochemical methods, their structure was established from the data of single crystals. There are two 1D polymers based on Ge8V12O48 cluster, and one 3D polymer with Ge6V15O48 structural unit. These studies were supplemented by the study of the catalytic activity of new and known similar compounds in styrene epoxidation processes.
This article may be of interest to readers of the journal due to its scientific component.
I have a few questions, the answers to which will allow me to publish this work.
1. The authors state that all synthesized compounds contain only vanadium(IV) ions. The source of vanadium is vanadium(V) oxide, there are no reducing agents in the reaction mixture. The only confirmation of paramagnetic ions in the obtained samples is the EPR spectra. EPR spectroscopy is the correct method for capturing unpaired electrons, some EPR spectra are shown in the SI, but they are not discussed. Could you clarify the number of unpaired electrons in the compounds under study? I advise you to confirm the oxidation state of vanadium by the second method, for example, electrochemistry, magnetochemistry, XPS. Distance analysis from X-ray data is insufficient for such a statement.
The degree of oxidation in new compounds is a controversial issue. For example, without taking into account the idea of the authors to protonate the Ge8V12O48 cluster, the calculation of the oxidation state of vanadium ions corresponds well to 5+. But the cluster will be neutral, which means that the presence of hydroxo groups in the outer sphere can be assumed. It can be assumed that 4 out of 12 vanadium ions have an oxidation state of 4+, then a balance will be maintained.
If you say that the cluster is protonated, then you should show this by measuring its charge in solution.
2. Please, check the structure of paper. I think it should contain separate chapters, Results and discussion (synthesis, X-ray data, catalytic activity) and Experimental part/Materials and Methods (methods, synthetic techniques (with the exception of 4 and 5, because a reference to the work where they are described is sufficient).
Round 2
Reviewer 1 Report
I agree the changes the authors have made and recommend publication of this manuscript in Molecules.
Reviewer 2 Report
I recommend accepting the article, all comments have been taken into account
This manuscript is a resubmission of an earlier submission. The following is a list of the peer review reports and author responses from that submission.